# GENERATIVE BLOCKS WORLD:
# MOVING THINGS AROUND IN PICTURES

**Vaibhav Vavilala**[1]  **Seemandhar Jain**[1]  **Rahul Vasanth**[1]  **D.A. Forsyth**[1]  **Anand Bhattad**[2]

[1]University of Illinois Urbana-Champaign  [2]Johns Hopkins University

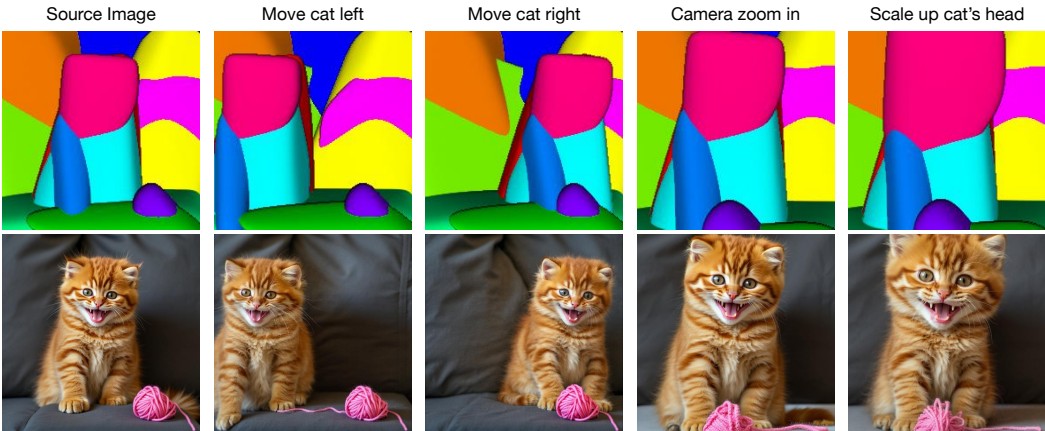

Figure 1: **Generative Blocks World.** Given an input image (bottom left), we extract a set of 3D convex primitives (top left) that provide an editable and controllable representation of the scene. These primitives are used to generate new images that respect geometry, texture, and the text prompt. The first column shows the original input and its primitive decomposition. Subsequent columns show sequential edits: translating the cat to the left (second column), translating it to the right (third column), moving the yarn in front of the cat and shifting the camera toward the scene center (fourth column), and scaling up the cat's head (burgundy primitive; fifth column). Our method enables 3D-aware semantic image editing through intuitive manipulation of these learned primitives.

## ABSTRACT

We describe Generative Blocks World to interact with the scene of a generated image by manipulating simple geometric abstractions. Our method represents scenes as assemblies of convex 3D primitives, and the same scene can be represented by different numbers of primitives, allowing an editor to move either whole structures or small details. Once the scene geometry has been edited, the image is generated by a flow-based method, which is conditioned on depth and a texture hint. Our texture hint takes into account the modified 3D primitives, exceeding the texture-consistency provided by existing techniques. These texture hints (a) allow accurate object and camera moves and (b) preserve the identity of objects. Our experiments demonstrate that our approach outperforms prior works in visual fidelity, editability, and compositional generalization.

## 1 INTRODUCTION

There is a rich literature treating editing real and generated images using various image-centered interfaces like dragging features. Interaction paradigms that exploit explicit representations of 3D are much less common. This paper describes an image editor built on a full 3D interaction paradigm, using a representation that is both compact and accurate – a *generative blocks world*.

Any scene representation that supports a **camera move** has some form of 3D representation. An explicit 3D representation helps, Fig 1. Explicit 3D representations have other important advantages. First, they offer **shape constancy**. When an object is moved across a perspective view, it is seen from a new aspect because the location of the focal point moves in object coordinates. This means

that (a) the shape of the object may change, with a change that depends on the field of view of the camera and the shape of the object and (b) some surface markings will become visible or invisible (e.g. bar code on soda can, Fig 3). When an object is moved toward or away from the camera, its image should expand or shrink (e.g. cat, Fig 1). If an editor does not preserve these properties correctly, the viewer may conclude that the shape or size of the object has changed. A properly constructed 3D representation will prevent this. Second, they offer **contact consistency**. A user who moves (say) a tin on a table generally expects the tin to remain in contact with the table. An explicit 3D representation allows the user to manage whether it does or not (e.g. dog in Fig 7; soda can, Fig 3). Third, they offer **shape completion**. Objects have backs that are not visible, but may have an effect when another object is moved in a scene. An explicit 3D representation can capture this effect.

It has been hard to build a 3D representation that: (a) represents the scene accurately enough that edited images are realistic and (b) is compact enough to support interactions. This paper uses modern fitting methods to represent scenes as small assemblies of meaningful parts or primitives (cf. *Blocks World* Roberts (1963) or *geons* Biederman (1987)). We call our method **Generative Blocks World**, *though our learned primitives are richer than cuboids*. Our method yields assemblies by decomposing an input image into a sparse set of convex polytopes (Vavilala et al., 2025a) that approximate the scene's depth map well enough to enable view-consistent texture projection. Further, our primitives respect object boundaries rather well. A user can reach into the scene and move a primitive, with predictable results. Our simple scene representations yield *hints* as to the appearance of the final image. These hints, together with the primitive depth map, are inputs to an off-the-shelf image generator, which renders accurate images.

Primitive decompositions have very attractive properties. They are *selectable*: individual primitives can be intuitively selected and manipulated (Fig. 1). They are *object-linked*: a segmentation by primitives is close to a segmentation by objects, meaning an editor is often able to move an object or part by moving a primitive (Figs 1; 3; 4). They are *accurate*: the depth map from a properly constructed primitive representation can be very close to the original depth map (Section 3.1), which means primitives can be used to build texture hints (Section 3.2) that support accurate camera moves (Figs 2; 5). They have *variable scale*: one can represent the same scene with different numbers of primitives, allowing multi-scale edits (Figs 7; 10; 13).

**Contributions:** **(1)** We describe a pipeline that fuses convex primitive abstractions with a flow-based generator to yield a natural 3D interaction paradigm for image editing. Our pipeline uses a texture-hint procedure that supports camera moves and edits at the object-level, while preserving identity. **(2)** We provide extensive evaluation demonstrating superior geometric control, texture retention, and edit flexibility relative to recent state-of-the-art baselines.

## 2 RELATED WORK

**Primitive Decomposition:** Early vision and graphics pursued parsimonious part-based descriptions, from Roberts' *Blocks World* Roberts (1963) and Binford's generalized cylinders Binford (1971) to Biederman's geons Biederman (1987). Efforts to apply similar reasoning to real-world imagery have been periodically revisited Gupta et al. (2010); Monnier et al. (2023); Bhattad et al. (2025) from various contexts and applications. Modern neural models revive this idea: BSP-Net Chen et al. (2020), CSG-Net Sharma et al. (2018), and CVXNet Deng et al. (2020) represent shapes as unions of convex polytopes, while Neural Parts Tulsiani et al. (2017), SPD Zou et al. (2018), and subsequent works Liu et al. (2022) learn adaptive primitive sets. Recent systems extend from objects to scenes: Convex Decomposition of Indoor Scenes (CDIS) Vavilala & Forsyth (2023) and its ensembling/Boolean refinement Vavilala et al. (2025a) fit CVXNet-like polytopes to RGB-D images, using a hybrid strategy. CubeDiff Kalischek et al. (2025) fits panoramas inside cuboids. Our work leverages CDIS as the backbone, but (i) improves robustness to in-the-wild depth/pose noise and (ii) couples the primitives to a Rectified Flow (RF) renderer, enabling controllable synthesis.

**Conditioned Image Synthesis:** Layout-to-image translation was pioneered in GANs Isola et al. (2017); Zhu et al. (2017); Park et al. (2019) and is now dominated by diffusion models such as Stable Diffusion Rombach et al. (2022), ControlNet Zhang et al. (2023), and T2I-Adapter Mou et al. (2024). These models can compose multiple spatial controls (Vavilala et al., 2024), perform

color edits (Vavilala et al., 2025b) and relight scenes Xing et al. (2025). We utilize a pretrained depth-conditional FLUX model, conditioning it on depth maps derived from our 3D primitives.

**Point-Based Interactive Manipulation:** Methods like DragGAN (Pan et al., 2023) and its diffusion-based successors (Shi et al., 2024; Mou et al., 2023; Cui et al., 2024; Pandey et al., 2024) offer intuitive 2D control by dragging handle points. Some approaches extend this to 3D using NeRFs for multi-view consistency Guang et al. (2025) or leverage self-guidance for layout control (Epstein et al., 2023). Our work differs by operating on editable 3D primitives instead of 2D points. This enables multi-resolution control and camera movement while handling perspective, occlusion, and texture.

**Object-Level and Scene-Level Editing:** Many recent works embed 3D priors for editing, though often focusing on single objects Gu et al. (2022); Wang et al. (2023); Poole et al. (2023); Tang et al. (2023); Cheng et al. (2025) or using language to guide transformations Michel et al. (2023). Our Generative Blocks World generalizes to complex edits not easily described by text. Another paradigm, seen in Image Sculpting Yenphraphai et al. (2024) and OMG3D Zhao et al. (2025), reconstructs an explicit 3D mesh for manipulation before re-rendering. While precise, these multi-stage pipelines can be bottlenecked by reconstruction quality. Our method provides a more streamlined approach by operating on abstract primitives, achieving strong geometric control without the complexity of direct mesh manipulation.

**Primitive-Based Scene Authoring:** LooseControl (Bhat et al., 2024) enables control via box-like primitives by fine-tuning a diffusion model with LoRA weights. This training is necessary to bridge the domain gap between its coarse primitive-based depth and standard depth maps (Yang et al., 2024). In contrast, our underlying primitive representation is accurate enough to require no fine-tuning. Furthermore, by abstracting objects into single, monolithic boxes, LooseControl is limited to holistic transformations and cannot perform part-level edits. Our method uses structured geometry, decomposing objects into multiple convex polytopes at variable levels of detail for more granular control. A more recent work, Build-A-Scene (Eldesokey & Wonka, 2025), uses a similar pipeline to LooseControl and thus inherits its limitations. Our approach differs by: (i) decomposing objects into multiple convex polytopes for finer control, (ii) supporting camera movement, and (iii) allowing novel scenes to be authored from scratch via primitive assembly.

## 3 METHOD

Generative Blocks World generates realistic images conditioned on a parsimonious and editable geometric representation of a scene: a set of convex primitives. The process consists of four main stages (Fig. 2): (i) primitive extraction from any image via convex decomposition (Sec. 3.1), (ii) generating an image conditioned on the primitives (and text prompt), (iii) user edits the primitives and/or camera, and (iv) generates a new image conditioned on the updated primitives, while preserving texture from the source image (Sec. 3.3). We describe each component in detail below.

### 3.1 CONVEX DECOMPOSITION FOR PRIMITIVE EXTRACTION

Our primitive vocabulary is blended 3D convex polytopes as described in Deng et al. (2020). CVXnet represents the union of convex polytopes using indicator functions $O(x) \to [0, 1]$ that identify whether a query point $x \in \mathbb{R}^3$ is inside or outside the shape. Each convex polytope is defined by a collection of half-planes.

A half-plane $H_h(x) = n_h \cdot x + d_h$ provides the signed distance from point $x$ to the $h$-th plane, where $n_h$ is the normal vector and $d_h$ is the offset parameter.

While the signed distance function (SDF) of any convex object can be computed as the maximum of the SDFs of its constituent planes, CVXnet uses a differentiable approximation. To facilitate gradient learning, instead of the hard maximum, the smooth LogSumExp function is employed to define the approximate SDF, $\Phi(x)$:

$$\Phi(x) = \text{LogSumExp}\{\delta H_h(x)\}$$

The signed distance function is then converted to an indicator function $C : \mathbb{R}^3 \to [0, 1]$ using: $C(x|\beta) = \text{Sigmoid}(-\sigma\Phi(x))$.

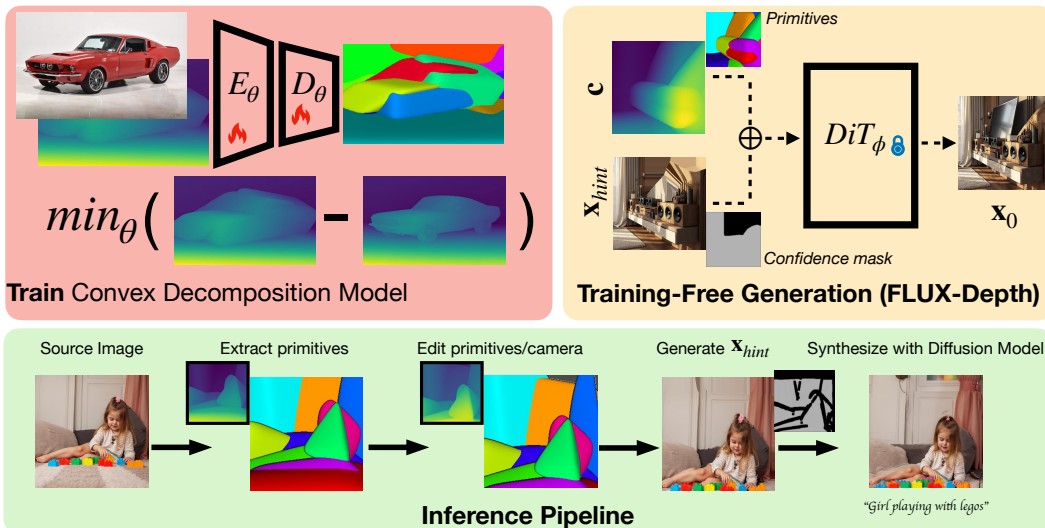

Figure 2: **Pipeline Overview. Top left:** We use convex decomposition models Vavilala et al. (2025a) to extract primitives from an input image at multiple scales. **Bottom:** Users can manipulate these primitives and the camera to define a new scene layout. We render the modified primitives into a depth map and generate a texture hint image. These serve as inputs to a pretrained depth-to-image model Labs (2024), which requires no fine-tuning (**Top right**). The generated image respects the modified geometry, preserves texture where possible, and remains aligned with the text prompt.

The collection of hyperplane parameters for a primitive is denoted as $h = \{(n_h, d_h)\}$, and the overall set of parameters for a convex as $\beta = [h, \sigma]$. While $\sigma$ is treated as a hyperparameter, the remaining parameters are learnable. The parameter $\delta$ controls the smoothness of the generated convex polytope, while $\sigma$ controls the sharpness of the indicator function transition. The soft classification boundary created by the sigmoid function facilitates training through differentiable optimization. For our primitive model we use ResNet-18 Encoder $E_\theta$ followed by 3 fully-connected layers that decode into the parameters of the primitives $D_\theta$. While the model is lightweight, the SOTA of primitive prediction requires a different trained model for each primitive count $K$.

Recent work has adapted primitive decomposition to real-world scenes (as opposed to well-defined, isolated objects, such as those in ShapeNet Vavilala & Forsyth (2023)). These methods combine neural prediction with post-training refinement: an encoder-decoder network predicts an initial set of convex polytopes, which is followed by gradient-based optimization to align the primitives closely to observed geometry. This approach is viable because the primary supervision for primitive fitting is a depth map (with heuristics that create 3D samples, and auxiliary losses to avoid degenerate solutions). Note that ground truth primitive parameters are not available (as they could be in many other computer vision settings e.g., segmentation Kirillov et al. (2023)). This is why the losses encourage the primitives to classify points near the depth map boundary correctly instead of directly predicting the parameters.

**Rendering the primitives.** We condition the RF model on the primitive representation via a depth map, obtained by ray-marching the SDF from the original viewpoint of the scene. Depth conditioning abstracts away potential 'chatter' in the primitive representation from e.g. over-segmentation, while simultaneously yielding flexibility in fine details (depth maps typically lack pixel-level high-frequency details). Depth-conditioned image synthesis models are well-established e.g. Zhang et al. (2023). Because **it's hard to edit a depth map, but easy to edit 3D primitives**, our work adds a new level of control to the existing image synthesis models. As we establish quantitatively in Table 2, our primitive generator is extremely accurate, and our evaluations show that we get very tight control over the synthesized image via our primitives. This means that whatever domain gap there is between depth from primitives and depth from SOTA depth estimation networks is not significant.

**Scaling to in-the-wild scenes.** We collect 1.8M images from LAION to train our primitive prediction models. To obtain ground truth depth supervision, we use DepthAnythingv2 Yang et al. (2024). We lift the depth map to a 3D point cloud using the pinhole camera model.

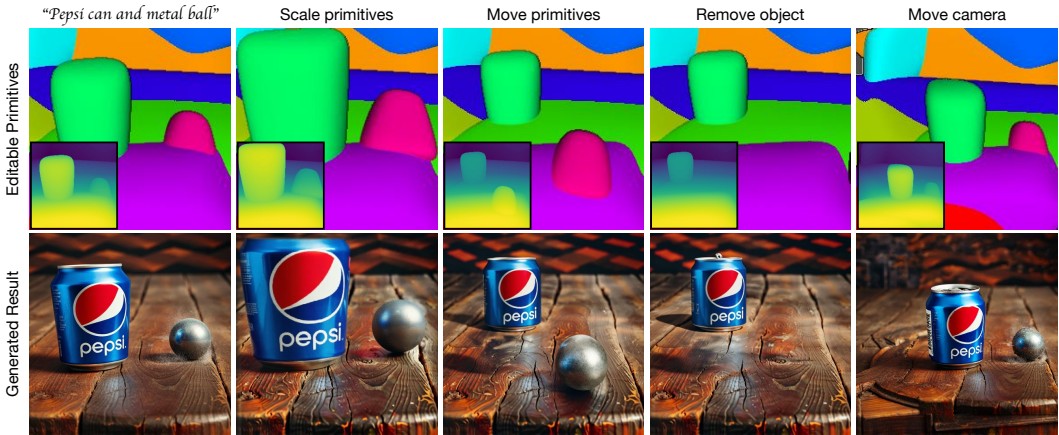

Figure 3: **Editable Primitives as a Structured Depth Prior for Generative Models.** Our method uses 3D convex primitives as an editable intermediate representation from which depth maps are derived. These depth maps (shown as insets in the top row) are used to condition a pretrained depth-to-image generative model. The top row shows primitive configurations after sequential edits—translation, scaling, deletion, and camera motion—alongside their corresponding derived depth maps. The bottom row shows the resulting synthesized images. Unlike direct depth editing, which is unintuitive and underconstrained, manipulating primitives offers a structured, interpretable, and geometry-aware interface for controllable image generation.

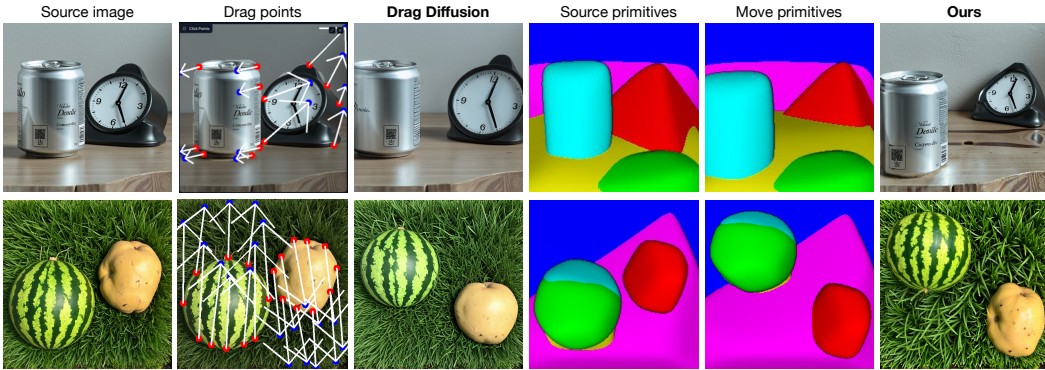

Figure 4: **Comparison with Drag Diffusion** (Shi et al., 2024). **First row:** Given a scene (first column), we attempt to reposition objects using a recent point-based image editing method by drawing drag handles (second column). However, drag points are ambiguous: it is unclear whether the intended operation is translation or scaling. As a result, the output lacks geometric consistency (third column). E.g., the clock changes shape, and pushing it deeper into the scene fails to reduce its size appropriately; fine details on the can are lost. In contrast, Generative Blocks World infers 3D primitives (fourth column) that can be explicitly manipulated (fifth column), producing a plausible image that respects object geometry, scale, positioning, and texture (last column). We also compare with proprietary models in supplement. **Second row** Drag Diffusion requires many arrows to place the objects. Notice how objects still do not move precisely where we want them, and there are shape and color mismatches on the rendered watermelon and potato. Our result respects both texture and geometry.

## 3.2 DEPTH-CONDITIONED INPAINTING IN RECTIFIED FLOW TRANSFORMERS

**Adding Spatial Conditions.** We build upon the state-of-the-art Flux, a rectified flow model Esser et al. (2024); Labs (2024). Older ControlNet implementations Zhang et al. (2023) train an auxiliary encoder that adds information to decoder layers of a base frozen U-Net. Newer implementations, including models supplied by the Black Forest Labs developers, concatenate the latent $x_t$ and condition (e.g., depth map) $c$ as an input to the network, yielding tighter control. `FLUX.1 Depth [dev]` re-trains the RF model with the added conditioning; `FLUX.1 Depth [dev] LoRA` trains LoRA layers on top of a frozen base RF model. Both options give tight control and

work well with our primitives, though LoRA exposes an added parameter $lora_{weight} \in [0, 1]$ tuning how tightly the depth map should influence synthesis. This is helpful when the primitive abstraction is too coarse relative to the geometric complexity of the desired scene (see Fig. 12).

**Role of Hint and Mask.** A core contribution of this work is an algorithm to generate a "hint" image to guide the image generation process, as well as a confidence mask (see Sec 3.3). The hint and mask influence the generation within timesteps $t_{end} \leq t \leq t_{start}$, which are hyperparameters. The mask $\mathbf{m} \in [0, 1]$ specifies regions where the hint should guide the output. The hint is encoded into latents $\mathbf{x}_{hint}$ via the VAE. During denoising, the latents are updated as $\mathbf{x}_t = (1 - \mathbf{m}) \cdot \mathbf{x}_{hint,t} + \mathbf{m} \cdot \mathbf{x}_t$, where $\mathbf{x}_{hint,t}$ is the noised hint latent at timestep $t$: $\mathbf{x}_{hint,t} = \text{SchedulerScaleNoise}(\mathbf{x}_{hint}, t, \boldsymbol{\epsilon})$. Thus, the hint image is *noised* to match the current timestep's noise level before incorporation, ensuring consistency with the denoising process. Outside $[t_{end}, t_{start}]$, the hint and mask are ignored.

### 3.3 TEXTURE HINT GENERATION FOR CAMERA AND OBJECT EDITS

A number of methods have been proposed to preserve texture/object identity upon editing an image. A common and simple technique is to copy the keys and values from a style image into the newly generated image (dubbed "style preserving edits"). For older U-Net-based systems, this is done in the bottleneck layers Bhat et al. (2024). For newer DiTs, this is done at selected "vital" layers Avrahami et al. (2025). In our testing, key-value copying methods are insufficient for camera/primitive moves (see Fig. 6). Further, because of our primitives, we have a geometric representation of the scene. Here we demonstrate a routine to obtain a source "hint" image $\mathbf{x}_{hint}$ as well as a confidence mask $\mathbf{m}$ that can be incorporated in the diffusion process. The hint image is a rough approximation of what the synthesized image should look like using known spatial correspondences between primitives in the first view and the second. The confidence mask indicates where we can and cannot trust the hint, commonly occurring near depth discontinuities. We rely on the diffusion machinery to essentially clean up the hint, filling gaps and refining blurry projected textures so it looks like a real image. The result of our process is an image that respects the text prompt, source texture, and newly edited primitives/camera.

**Creating point cloud correspondences** We develop a method that accepts point clouds at the ray-primitive intersection points, a *convex_map* integer array indicating which primitive was hit at each pixel, a list of per-primitive transforms (such as scale, rotate, translate), and a hyperparameter $max\_distance$ for discarding correspondences. This procedure also robustly handles camera moves because the input point clouds are representations of the same scene in world space.

**Creating a texture hint** Given a correspondence map of each 3D point in the new view relative to the original view, we can apply this correspondence to generate a hint image that essentially projects pixels in the old view onto the new view. This is the $\mathbf{x}_{hint}$ supplied to the image generation model, taking into account both camera moves and primitive edits like rotation, translation, and scaling. The point cloud correspondence ensures that if a primitive moves, its texture moves with it. In practice, this hint is essential for good texture preservation (see Fig. 6). Correspondence and hint generation take about 1-2 seconds per image; 30 denoising steps of FLUX at 512 resolution take about 3 seconds on an H100 GPU.

### 3.4 EVALUATION

We seek error metrics to establish (1) geometric consistency between the primitives requested vs. the image that was synthesized and (2) texture consistency between the source and edited image. For (1) we compute the AbsRel between the depth map supplied to the depth-to-image model (obtained by rendering the primitives) and the estimated depth of the synthesized image (we use the hypersim metric depth module from Yang et al. (2024) to get linear depth). Consistent with standard practice in depth estimation, we use least squares to fit scale and shift parameters onto the depth from RGB (letting the primitive depth supplied to the DM be GT).

To evaluate texture consistency, we apply ideas from the novel view synthesis literature and our existing point cloud correspondence pipeline. Given the source RGB image and the synthesized RGB image (conditioned on the texture hint), we warp the second image back into the first image's frame using our point cloud correspondence algorithm. If we were to synthesize an image in the first render's viewpoint using the second render, this is the texture hint we would use. In error met-

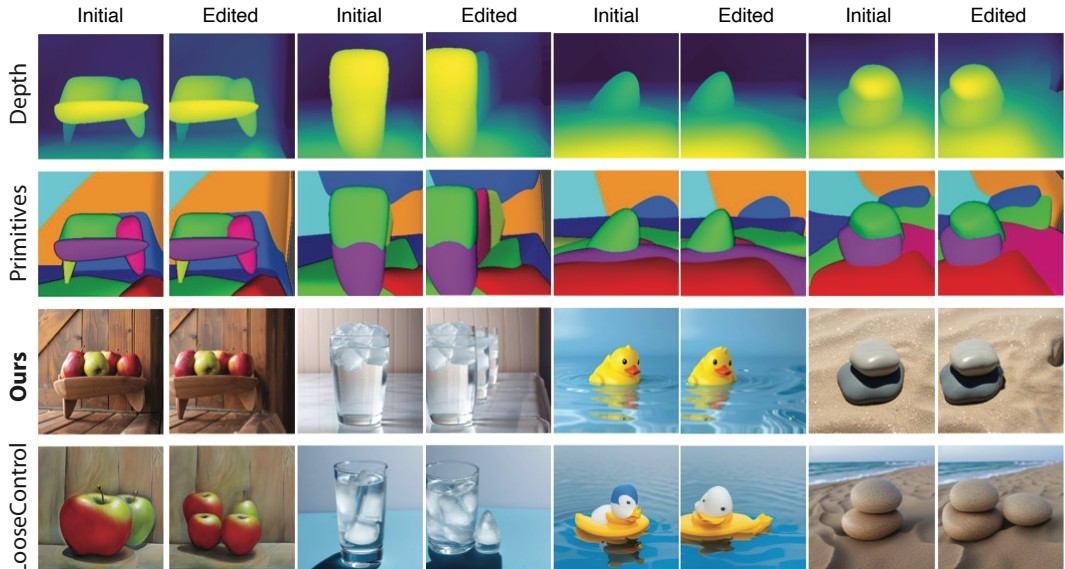

Figure 5: **Comparison with LooseControl** Bhat et al. (2024). Existing work struggles with camera moves. Four scenes (**left** side of each pair), synthesized from the depth maps shown. In each case, the camera is moved to the right (**right** side of each pair), and the image is resynthesized. Note how, for LooseControl, the number of apples changes (first pair); the level of water in the glass changes and there is an extra ice cube (second pair); the duck changes (third pair); and an extra rock appears (fourth pair). In each case, our method shows the same scene from a different view, because the texture hint image is derived from the underlying geometry, and strongly constrains any change.

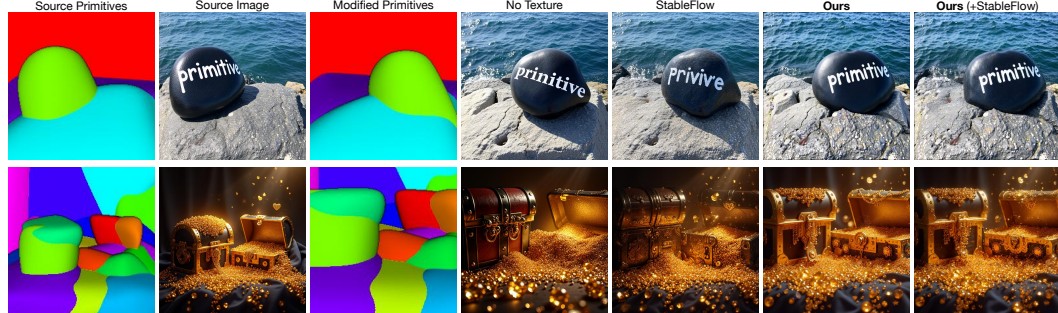

Figure 6: **Projection-Based Texture Hints Preserve Object Identity After Edits.** This figure compares our projection-based texture hints against StableFlow Avrahami et al. (2025), which uses vital-layer key-value injection. **First two columns:** input primitives and image. **Third:** edited primitives. **Fourth:** synthesis from original depth, revealing consistent geometry but altered texture. **Fifth:** StableFlow's approach often changes texture or object identity. **Sixth:** our projection-based hints maintain texture fidelity despite edits. **Seventh:** combining both approaches sometimes improves fine detail recovery (e.g., the treasure chest).

ric calculation, the first RGB image is considered ground truth, the warped RGB image from the edited synthesized image is the prediction, and the confidence mask filters out pixels that are not visible in view 1, given view 2. This evaluation procedure falls in the category of cycle consistency/photometric losses that estimate reprojection error Fang et al. (2024); Jeong et al. (2024); Li et al. (2025); Qin et al. (2025).

## 4 RESULTS

Fig. 4 shows how users can manipulate depth map inputs to depth-to-image synthesizers; Fig 5 shows camera moves. We have precise control over synthesized geometry while respecting texture. The evaluation in Table 1, demonstrates we hit both goals conclusively. Existing texture preserva-

Table 1: Comparison of image reconstruction and generation metrics between our method and LooseControl. **AbsRel**src and **AbsRel**dst are absolute relative errors evaluating how well the generated images adhere to the requested primitive geometry (source and modified, respectively). PSNR and SSIM are evaluated by reprojecting the second synthesized image back to the original camera viewpoint (see Sec 3.4) and measuring texture consistency with the source. Observe how our procedure simultaneously offers tight geometric adherence to the primitives while preserving the source texture. Results obtained by averaging 48 test images with random camera moves. Because Bhat et al. (2024) does not offer primitive extraction code, we supply our own primitives to both methods for evaluation. We use $K = 10$ parts for this evaluation.

| Method | AbsRel$_{src}$ ↓ | AbsRel$_{dst}$ ↓ | PSNR ↑ | SSIM ↑ |
|---|---|---|---|---|
| **Ours** | **0.072** | **0.076** | **18.7** | **0.874** |
| LooseControl Bhat et al. (2024) | 0.143 | 0.146 | 6.65 | 0.670 |

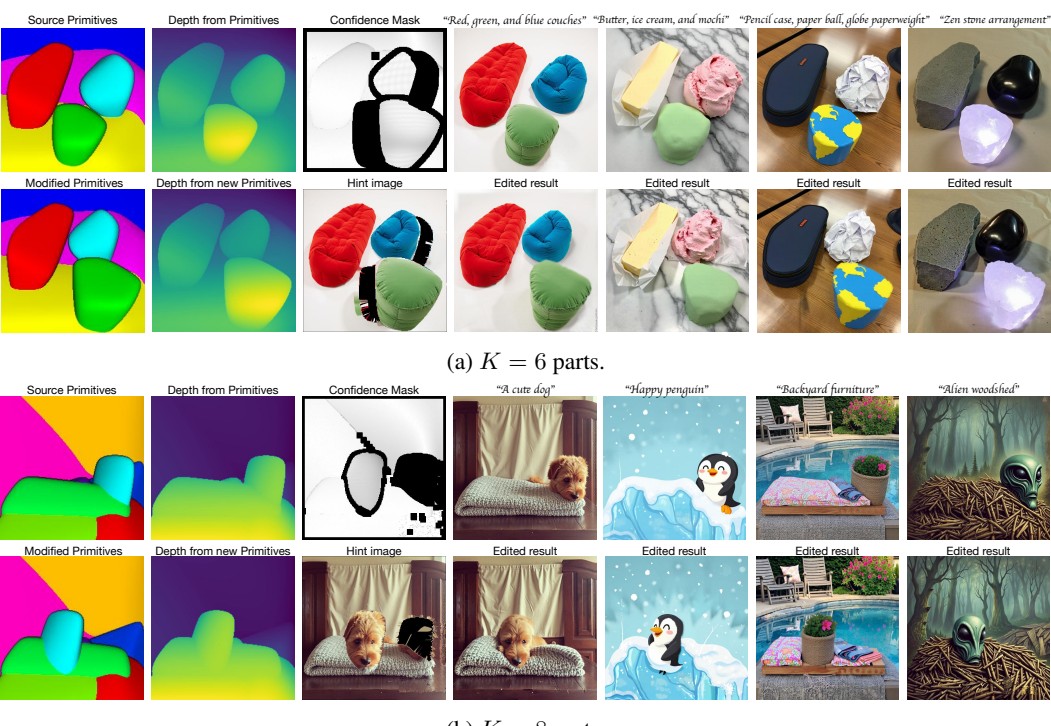

(a) $K = 6$ parts.

(b) $K = 8$ parts.

Figure 7: **Applying same primitive edit for different text prompts at coarse scale ($K \in \{6, 8\}$ parts)**. First row in each subplot contains source primitives and depth (first two columns); the confidence mask for hint generation, followed by four source RGB images. Second row shows the modified primitives and depth, followed by the hint image $x_{hint}$, followed by the four corresponding edited images. At coarse scales, moving a primitive can move a lot of texture at once. Observe how our hint generation procedure automatically yields confidence masks and hints, assigning low confidence to boundaries of primitives that moved (e.g., the dog's hair) and reveals holes when moving objects. The image model cleans up the low-confidence regions and even handles blurry/aliased texture in the hint when $t_{end} > 0$, meaning that the hint is not used for some denoising steps.

tion based on key-value transfer do not preserve details very well, only high-level semantics and style. We ablate the advantage of our texture preservation approach in Fig. 6. When there are few primitives, moving one primitive affects a big part of the scene; when there are a lot of primitives, we can make fine-scale edits. We show several such examples in Figs. 7, 10 in supplement.

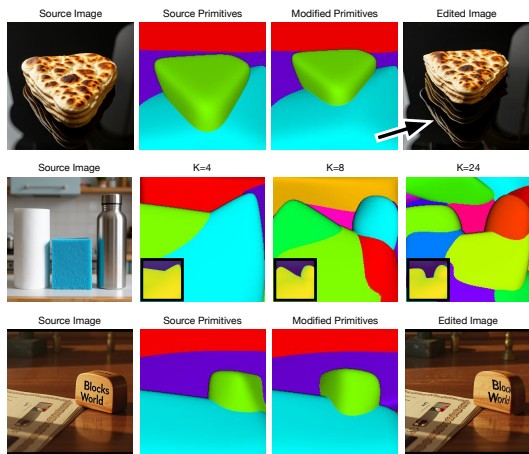

Figure 8: **Failure cases. Top: Illumination misalignments.** Our pixel-space texture hints fail to model lighting (e.g., reflections, shadows) outside primitive boundaries. Consequently, moving an object like the bread stack does not update its static reflection. **Middle: Poor decomposition.** In cluttered scenes or near image edges, sparse depth can cause primitive fitting to fail, incorrectly merging adjacent objects (bottle and paper towel) and resulting in poor control. **Bottom: Rotation artifacts.** Large object rotations (50 degrees) disrupt geometry and texture consistency, causing distortions or hallucinated content (warped text), likely due to a distribution shift in the texture hints.

## 5 DISCUSSION

3D primitives offer precise geometric control over image generation model outputs, and preserve high-level textures more effectively than key-value transfer methods. Our method works because primitive decompositions offer several useful properties: they are selectable; they are object-linked; they are compact; they allow edits at coarse and fine grain; and they are accurate enough to yield depth maps that support high-quality texture projection. Our pipeline is designed to allow users to choose between coarse and fine control by adjusting the number of primitives to suit the editing task and scene context.

Our methods have difficulty with some non-convex shapes (e.g. underside of a chair or handle of a coffee mug); additional segmentation and masking, more primitives, or more types of primitive might help. Depth-of-field blurring/bokeh may not be resolved or sharpened when bringing out-of-focus objects into focus. Significant object rotations may also fail (see Fig. 8). In an interactive workflow, manually expanding the confidence mask to include problematic regions e.g., unwanted reflections that don't move with a primitive, can fix some issues. Future work that applies our point correspondences within the network layers themselves (e.g., in vital layers) may yield more robust solutions. Our method does not account for view-dependent lighting effects and does not enforce temporal consistency across frames for video synthesis.

Our work highlights the delicate links between the text prompt, hint image, initial noise tensor, and depth map. Current inverters do not support our editing model, apparently because edited images should start from the same noise tensor and prompt as the source image to achieve good results. Certain edits that are at odds with the text prompt are likely to cause problems (e.g., if the prompt mentions an object is on the right, but a user manipulates the primitives to move the object to the left). Changing the text prompt could work in some circumstances (Fig. 11).

ACKNOWLEDGEMENT

This material is based upon work supported by the National Science Foundation under Grant No. 2106825. This research used both the DeltaAI advanced computing and data resource, which is supported by the National Science Foundation (award OAC 2320345) and the State of Illinois, and the Delta advanced computing and data resource which is supported by the National Science Foundation (award OAC 2005572) and the State of Illinois. Delta and DeltaAI are joint efforts of the University of Illinois Urbana-Champaign and its National Center for Supercomputing Applications.

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
