# OpenReview forum: "Generative Blocks World: Moving Things Around in Pictures"
_ICLR.cc/2026/Conference — ICLR 2026 Poster_

### Official Review · Reviewer_Ka9k · 2025-10-28

**Soundness:** 3
**Presentation:** 2
**Contribution:** 2
**Rating:** 4
**Confidence:** 3

**Summary:**

This paper proposes a new system named Generative Blocks World, for interactive image editing. The authors argue that primitive is a more straightforward, and intuitive concept for user to interact with rather than raw depth map. The detailed process of the model includes a primitive extraction model, hint image generation process and final image editing model. The authors first train the primitive extraction model on a subset of LAION dataset to predict primitives (with known number for each trained model). Then given an image to be edited, user can manipulate the extracted primitives to achieve both camera view editing and scene components editing. The edited primitives are used to generate hints: including depth map, and a "warped" hint image. These components are used to condition a depth-conditioned FLUX model for final output. The authors also conduct extensive analysis of the proposed system to show the validness of each component.

**Strengths:**

The proposed Generative Blocks World has a good motivation. Primitive is a more straightforward concept comparing to raw depth map, and varying number of primitives also provide a flexible way to understand the image content.

Authors also conduct thorough analysis on the proposed method's components, which provides a very detailed understanding.

The results look promising with both camera control and image component control.

**Weaknesses:**

The most important weakness of this paper is currently I think the authors are lack of analysis of the importance of the proposed "primitives". From my perspective, the actual image generation model is a finetuned FLUX model that conditions on a combination of "hint image, depth, (mask?)" The authors should:
1. Validate the effectiveness of each components in the condition combination through some qualitative examples or analysis. But I also understand if this requires retrain the model and time is limited. But I think at least some intuitive discussion is needed.
2. This pipeline, in my opinion, can actually be achieved via: use segmentation mask to achieve the same usage of primitives, and use SDEdit + a depth-conditioned FLUX model to accept both hint image and depth control. So I think authors should do some qualitative comparisons with this one or even three baselines to prove the validness of the proposed Generative Block World:
- move segmentation mask to warp the depth and hint image, still use current (finetuned?) FLUX-Depth
- primitive pipeline as current, but use original FLUX-Depth + SDEdit for hint image condition
- move segmentation mask to warp the depth and hint image, use original FLUX-Depth + SDEdit for hint image condition
I think these baselines are valid, but maybe time is limited so some comparison examples would be fine

**Questions:**

I feel like the paper is written in a rush and some details are not clear, so maybe my aforementioned weakness is with some misunderstandings.
My questions include:
1. did the authors actually fine-tune the FLUX-Depth to take hint image as input, or actually not, then that step is just SDEdit? (Because it seems very much like SDEdit)
2. What exactly are the conditions from primitive manipulation? Depth? Hint image? What about the confidence mask, I think this part requires an additional figure to explain for better understanding.
3. What's the training time consumption for the primitive extraction model? And if there is, what's the training consumption for FLUX-Depth finetune? Is there any difference in training dynamics for primitive extraction model with different primitive numbers?
4. See weakness above.

---

> ### Author Response · Authors · 2025-12-03
>
> Thank you for your helpful and detailed review of our paper!
>
> 1. Training-Free. FLUX-Depth is pretrained, not finetuned by us. Our contribution is developing 3D primitives accurate enough such that the depth-to-image model doesn’t need to be finetuned, as well as the generation of texture hints. Specifically, we only train primitive prediction models and do not update any weights of the FLUX generator.
>
> 2. Ablations. Please see the rebuttal pdf for additional ablations on the role of the hint/confidence mask, diffusion timestep window to apply the mask, dilation hyperparameter, and lora weight. All of these are now quantitatively and qualitatively ablated.
>
> 3. On Segmentation idea. While we appreciate the suggestion regarding 2D segmentation, we must reiterate that our core objective is to enable complex 3D edits. When a user specifies camera transforms or manipulates primitive-scale/rotate/translate parameters, the underlying model must inherently handle occluded regions, maintain 3D awareness, correctly infer how object size changes with depth, and how to deal with extrapolation of new pixels. These are operations that are straightforward in a 3D interface (which then projects to 2D depth) but are fundamentally impossible using only 2D segmentation. Specifically, 2D segmentation cannot resolve scale ambiguity (how much an object shrinks when moved deep into the scene; for example, the clock in Fig 4) or provide the necessary 3D awareness for these transformations (the upper tin cover that becomes visible on the pepsi can in the last two rows in Fig 3 as well as the bar code that was invisible earlier is added). Such edits are challenging to obtain with only 2D segments.
>
>
> However, we agree that combining segment boundaries with our 3D primitives could potentially refine the precision of our texture hints. Building a full system that uses segmentation without our primitives to infer 3D structure for movement, as you propose, appears to be an overly complex baseline.
>
> The key insight of our work is this: although depth-to-image models are highly developed, editing the depth map accurately remains extremely difficult without a robust 3D decomposition method, which our primitives provide. We believe this lack of a practical, 3D-aware editing approach based on primitives represents a significant gap in the literature, which our proposed method is designed to fill.

---

> > ### Author Response · Authors · 2025-12-03
> > **Questions**
> >
> > 1. We did not finetune FLUX-depth; we use a pretrained model. Our primitives give depth guidance. Our texture hint and confidence mask influence the synthesis (see our methodology). Thus, the final conditions are: text prompt, depth map, confidence mask, and hint.
> >
> > 2. See (1) above, and Fig. 2 from the paper. We will lightly polish this figure to make the conditions clearer.
> >
> > 3. In the Appendix line 719-729, we state the training time for the primitive models: 40-100 minutes per model, depending on the number of primitives. As we mentioned, there is no finetuning of FLUX.

---

### Official Review · Reviewer_VvNT · 2025-10-29

**Soundness:** 3
**Presentation:** 2
**Contribution:** 3
**Rating:** 4
**Confidence:** 3

**Summary:**

This paper introduces "Generative Blocks World," a novel and interesting framework for image editing that leverages explicit 3D representations. The core idea is to decompose an image into a set of 3D convex primitives, which a user can directly manipulate (e.g., move, scale, rotate) to edit the scene's geometry. A new image is then synthesized by a pretrained, depth-conditioned generative model (FLUX), guided by a depth map rendered from the edited primitives and a novel texture hint mechanism designed to preserve object identity. The motivation is clear, and the proposed method offers a training-free (at inference time) approach to solving challenging, geometry-aware image editing tasks.

**Strengths:**

1. Novelty and Motivation: The approach of revitalizing classic "blocks world" concepts for controlling modern generative models is highly innovative. It provides a clear and compelling solution for 3D-aware image manipulation, which is a significant problem in the field.

2. Decoupling of Geometry and Texture: The framework effectively decouples geometric control (via primitives and depth maps) from appearance generation (via the generative model and texture hints). This modularity is a key strength, allowing for precise and predictable geometric edits.

**Weaknesses:**

1. Limited Quantitative Evaluation: The quantitative comparison is confined to a single baseline (LooseControl) on a small, unstated set of test images. The paper's claims of superiority would be significantly strengthened by a more extensive evaluation on standard image editing benchmarks and against a wider array of recent methods, especially those with different interaction paradigms (e.g., drag-based).

2. Lack of Critical Ablation Studies: The paper is missing important ablation studies that would provide valuable insight into the method's internal mechanics and sensitivity to hyperparameters. For instance, an analysis of how the texture hint's application window (i.e., the start and end timesteps, $t_{start}$ and $t_{end}$) affects generation quality is crucial. Without such analysis, the reader cannot fully appreciate the contribution of this specific component or understand the trade-offs involved in its tuning.

3. Unaddressed User Interaction Challenges: The paper claims that manipulating primitives is an "intuitive" and "natural"  interaction paradigm. However, it does not address the significant practical challenges of this interface. For complex scenes or objects decomposed into many primitives, requiring the user to manually select, group, and manipulate these components could be tedious and imprecise. The paper would benefit from a more detailed discussion of the proposed interaction workflow and, ideally, a user study to validate its usability claims against other interfaces.

4. Unintended Background Alterations: A notable weakness, stemming from the reliance on a holistic generative model like FLUX, is the tendency for static background regions to change during an edit. When a foreground object is manipulated via the depth map, the model re-synthesizes the entire image to ensure global coherence. This often results in undesirable and unpredictable alterations to the background, which should have remained unchanged. This lack of localized control undermines the precision that the primitive-based editing promises.

**Questions:**

Same as weakness.

---

> ### Author Response · Authors · 2025-12-03
>
> Thank you for reviewing our paper and recognizing its novelty and modularity.
>
> 1. We added more quantitative evaluation in Table 4, Fig. 18, Fig. 20, and Fig. 23 on the open DragBench dataset. As mentioned in the global comment, there isn’t a standard way to compare drawing arrows (drag-based) with specifying camera/primitive 3D parameters (ours). However, in addition to Fig. 4 of the main text, our rebuttal pdf includes two more qualitative examples demonstrating the types of edits our method excels at, but drag-based struggles with (Fig. 22).
>
> 2. Please see the rebuttal pdf for additional ablations on the role of the hint/confidence mask, diffusion timestep window to apply the mask, dilation hyperparameter, and lora weight. All of these are now quantitatively and qualitatively ablated.
>
> 3. This is a valuable discussion, and we will add more details to the main text and commit to open source our full pipeline. Designing a production-ready UI is an engineering challenge, not a primary research outcome, and making our prototype UI available to enough people for comparison with, say, drag-based UIs was not feasible in the given timeframe. Instead, we supplied a large number of evaluations using frontier methods to illustrate the types of edits that are easy with our approach but challenging with others. Our key research goal was to investigate whether simple 3D primitive edits, when fed to a standard depth-conditioned generator, could result in desired outcomes, which our results affirm. The core interaction mechanism for our approach is moving a 3D blob or camera in a UI (there are many such available in 3D editing software like Autodesk Maya, though we built our own using Streamlit for this project). Additionally, users can manually specify per primitive (and camera) scale/rotate/translate parameters for precise control. We haven’t seen this level of control in existing work, and believe ours should be part of the rich family of modern image editing methods.
>
> 4. This is a great point - many image editing methods alter the background in an unintended way (for example, see DragDiffusion result on the dog in Fig. 22). The main objective of our hint mechanism is to preserve this texture, and that is precisely what we evaluate in Table 2 (main text) and Table 4 in our rebuttal. We reproject texture from the new view back into the original view and measure texture consistency. Ours compares favorably with the baselines, and in particular, we supplied several qualitative images for the reader to get a sense of how our texture preservation compares with existing work, including frontier models. We believe ours is very competitive.

---

### Official Review · Reviewer_6yrZ · 2025-10-31

**Soundness:** 2
**Presentation:** 3
**Contribution:** 2
**Rating:** 4
**Confidence:** 3

**Summary:**

Learn to do controlled spatial edits to a photo by learning to extract editable primitive 3D representations of the image contents which can then be modified by a human easily and conditioned on do generate a new view of the modified scene. Core idea is to find a way of doing 3D modifications to an image that is easier to do for a human and condition a diffusion model on.

**Strengths:**

Lots of qualitative demos for the paper and the quality of the model at preserving scene contents.

**Weaknesses:**

Very minimal quantitative results to compare with other works, quantitative results feel like they're lacking overall. There are other works that do a similar style of task prompting from optical flow / correspondences (motion prompting and go-with-the-flow were the ones I knew, but you also referenced drag-diffusion). I might be wrong, but it feels like some more quantitative comparisons could be done against these kinds of models perhaps? This paper is an interesting way of approaching the problem of image edits but as someone that isn't working on this specific problem, I just don't have enough of an idea of how this sits in the landscape of existing works from the quantitative results provided.

**Questions:**

Is there any reason why more quantitative baselines cannot be run to better inform the audience of how this work sits in comparison to other existing works?

---

> ### Author Response · Authors · 2025-12-03
>
> Thank you for reading our paper and the helpful feedback! We added more quantitative evaluation in Table 4, Fig. 18, Fig. 20, and Fig. 23. These evaluations ablate core components of our new method (e.g. the confidence mask mechanism and hyperparameters), as well as existing image editing techniques (Loose Control, StableFlow).
>
> We added Fig. 22 to specifically compare with DragDiffusion; Figs. 4, 5, 6, and 9 also serve as critical evaluations against key works in this space, including frontier models. The key theme is that our method can achieve edits that other works struggle with.

---

### Official Review · Reviewer_UHGg · 2025-11-03

**Soundness:** 4
**Presentation:** 3
**Contribution:** 3
**Rating:** 8
**Confidence:** 4

**Summary:**

This paper introduces a method for 3d aware image editing. This method directly edits 3D primitives, which is very accurate, and more intuitive to interact. The method can be divided into these stages: 1) decompose 2d image into 3d primitives, using pretrained convex decomposition models 2) edit the 3d primitives 3) render the modified primitives into depth map and texuture hint image 4) use a diffusion model Flux Depth to convert the depth map and texture hint image into the final edited image

**Strengths:**

1. This method is training free, by leveraging existing models, it is able to provide a very good image editing result
2. The speed is very fast, better than other 3d aware editing methods I known
3. The texture Hint injection is pretty good, do not require training the diffusion model, just doing the injection during inference, but still get a very good result. It even keeps the text texture, which is amazing.
4. Overall, I love this paper very much, it leverages foundamental graphics techniques, but achieving something big! The quality is very good, and the methods offer a great speed.
5. The failure cases are fully discussed

**Weaknesses:**

Thanks a lot for the authors taking a time to discuss the failure cases, I feel the discussions are very valuable. All the weaknesses are acceptable, and I feels some can be solved by more advanced models within this framework. For example, the first row in Figure 8 is very likely to be a failure of Flux Depth, not the framework itself.

**Questions:**

1. Is it possible to show some hint images for text textures, like the cases in row 3 of Figure 8, I feel this will helps us to better understand the texture mismatch.

---

> ### Author Response · Authors · 2025-12-03
>
> We're grateful for the reviewer’s time and for recognizing our contributions. Please see Fig. 25 for the hint image that the reviewer has requested. This is on the final page of the rebuttal pdf (that we attached in the supplement pdf location).

---

### Author Response · Authors · 2025-12-03
**Global comment**

We thank the reviewers and ACs for their time in reading our manuscript. We first highlight the enthusiastic consensus on the paper's impact. Reviewer UHGg encapsulated this: "Overall, I love this paper very much... it leverages fundamental graphics techniques, but achieving something big!" VvNT praised the "highly innovative" approach of "revitalizing classic 'blocks world' concepts," which enables a "training-free" framework where "the speed is very fast" (UHGg). This efficiency does not compromise quality; 6yrZ lauded the "quality of the model at preserving scene contents," while Ka9k confirmed "results look promising with both camera control and image component control." VvNT further emphasized that decoupling geometry and texture is a "key strength," and UHGg appreciated that "failure cases are fully discussed," demonstrating scientific rigor.

To summarize, our contribution brings a classic image representation - 3D primitives - into modern relevance via controlled image synthesis. In particular, we build a new pipeline to use 3D primitives to generate texture-consistent and geometrically accurate images. In our work, we train a primitive prediction model that fits primitives to RGB images (obtaining a 3D reconstruction). We then use these primitives as part of a training-free framework to condition a FLUX-depth diffusion model to generate new images that respect the desired 3D primitive edits. We will open-source all our code and models.

Please see rebuttal.pdf (in the supplement pdf attachment location) for our rebuttal response (which we will integrate with the main paper). We respond to each reviewer’s concerns and questions below. We want to address one major point raised by the reviewers.
Several reviewers requested an evaluation against drag-based methods. See Fig. 4 in the main text and Fig. 22 in the rebuttal for qualitative comparisons. Quantitative evaluation of drag-based methods is not feasible due to the fundamental differences in interaction paradigms. Drag-based approaches require users to draw arrows, which often become complex and dense even for the basic edits in our paper. For example, to move something without deforming it requires that the arrows represent a rigid motion flow field, which can be tricky for users (see Figure 22 of rebuttal).  Our method uses camera translation parameters and scale, rotate, and translate controls for each primitive, which drag-based approaches do not support. This makes a fair, apples-to-apples quantitative comparison impossible. Drag-based approaches simply are not designed for the type of edits our method enables.
Moreover, quantitative evaluation with drag-based methods is also not scalable. Achieving the controlled edits shown in our work would demand extensive manual drawing and dragging, making large-scale evaluation impractical. Instead, we provide substantial quantitative results using 3D geometry (box) edits as a strong baseline. Our initial submission also included qualitative comparisons with state-of-the-art proprietary models, including OpenAI ChatGPT, Gemini Nano Banana, and Reve (see Fig. 9). These models also struggle to handle the 3D edits that our method supports.
Notably, drag-based methods lack 3D awareness, while our edits are grounded in 3D space because of the 3D convex decomposition of the entire scene. Our qualitative results show that drag-based methods struggle with basic 2D translation tasks, often resulting in scale ambiguity and undesired texture changes. The provided figures highlight the strengths and unique features of our approach -- offering a robust and geometrically intuitive framework for image editing that addresses the limitations of drag-based approaches and enables precise and controlled image synthesis.

---

### Meta-Review · Area_Chair_x5Dw · 2026-01-07

**Summary:**

The paper proposes "Generative Blocks World," a novel training-free framework for controllable image editing. The method decomposes images into 3D convex primitives ("blocks"), allowing users to manipulate scene geometry (translation, rotation, scale) before regenerating the image using a depth-conditioned FLUX model and a texture hint mechanism. Reviewer UHGg strongly championed the paper, praising its innovation in "revitalizing classic blocks world concepts" and its "training-free" efficiency. While other reviewers initially rated it marginally below the threshold due to a lack of quantitative comparisons, they universally acknowledged the clear motivation, the effective decoupling of geometry and texture, and the promising qualitative results. Also the authors did a detailed rebuttal to address the concerns.

**Reviewer Concerns:**

Reviewer Concerns addressed by the rebuttal:

1. Quantitative Baselines and Drag-based Comparisons (Reviewers 6yrZ, VvNT): The authors added extensive quantitative evaluations (Table 4, Figs. 18, 20, 23), including performance on the DragBench dataset.
2. Ablation Studies (Reviewers VvNT, Ka9k): The authors included new qualitative and quantitative ablations covering the hint/confidence mask, diffusion timestep windows, dilation hyperparameters, and LoRA weights in the rebuttal PDF.
3. Justification of 3D Primitives vs. 2D Masks (Reviewer Ka9k): The authors persuasively argued that 2D segmentation cannot resolve scale ambiguity (object shrinking with depth) or handle occlusion and extrapolation during 3D manipulation.

Reviewer Concerns not addressed:

Reviewer VvNT raised a valid practical point regarding User Interaction Challenges, noting that selecting and manipulating many primitives in complex scenes could be tedious and imprecise.

**Reviewer Scores:**

**Reviewer UHGg (Score: 8 -> 8):** The reviewer was already highly enthusiastic, calling the paper "excellent" and stating they "love this paper very much".

**Reviewer 6yrZ (Score: 4 -> 6):** This reviewer's primary concern was the "very minimal quantitative results" and lack of comparisons to drag-diffusion. The authors directly addressed this by adding quantitative evaluations, which would likely raise the score to an Accept.

**Reviewer VvNT (Score: 4 -> 4):** Concerns regarding "limited quantitative evaluation" and "lack of critical ablation studies" were the main drawbacks cited. The rebuttal provided these missing ablations and quantitative data, however, the UI interaction challenge may still exist so the score may be unchanged.

**Reviewer Ka9k (Score: 4 -> 6):** This reviewer questioned the necessity of 3D primitives over 2D segmentation and whether the model was fine-tuned. The rebuttal clarified the model is training-free and explained why 2D segmentation fails at scale ambiguity/occlusion where 3D primitives succeed.

---

### Decision · Program_Chairs · 2026-01-26

Accept (Poster)